# Microalgal Derivatives as Potential Nutraceutical and Food Supplements for Human Health: A Focus on Cancer Prevention and Interception

**DOI:** 10.3390/nu11061226

**Published:** 2019-05-29

**Authors:** Christian Galasso, Antonio Gentile, Ida Orefice, Adrianna Ianora, Antonino Bruno, Douglas M. Noonan, Clementina Sansone, Adriana Albini, Christophe Brunet

**Affiliations:** 1Stazione Zoologica Anton Dohrn, Villa Comunale, 80121 Naples, Italy; christian.galasso@szn.it (C.G.); antonio.gentile@szn.it (A.G.); ida.orefice@szn.it (I.O.); adrianna.ianora@szn.it (A.I.); christophe.brunet@szn.it (C.B.); 2Laboratory of Vascular Biology and Angiogenesis, IRCCS MultiMedica, 20138 Milan, Italy; 82antonino.bruno@gmail.com (A.B.); douglas.noonan@uninsubria.it (D.M.N.); 3Department of Biotechnology and Life Sciences, University of Insubria, 211000 Varese, Italy; 4School of Medicine and Surgery, University of Milano-Bicocca, 20900 Monza, Italy

**Keywords:** functional food, microalgae, nutraceutical, chemoprevention, marine bioactive compounds, cancer

## Abstract

Epidemiological studies are providing strong evidence on beneficial health effects from dietary measures, leading scientists to actively investigate which foods and which specific agents in the diet can prevent diseases. Public health officers and medical experts should collaborate toward the design of disease prevention diets for nutritional intervention. Functional foods are emerging as an instrument for dietary intervention in disease prevention. Functional food products are technologically developed ingredients with specific health benefits. Among promising sources of functional foods and chemopreventive diets of interest, microalgae are gaining worldwide attention, based on their richness in high-value products, including carotenoids, proteins, vitamins, essential amino acids, omega-rich oils and, in general, anti-inflammatory and antioxidant compounds. Beneficial effects of microalgae on human health and/or wellness could in the future be useful in preventing or delaying the onset of cancer and cardiovascular diseases. During the past decades, microalgal biomass was predominately used in the health food market, with more than 75% of the annual microalgal biomass production being employed for the manufacture of powders, tablets, capsules or pastilles. In this review, we report and discuss the present and future role of microalgae as marine sources of functional foods/beverages for human wellbeing, focusing on perspectives in chemoprevention. We dissected this topic by analyzing the different classes of microalgal compounds with health outputs (based on their potential chemoprevention activities), the biodiversity of microalgal species and how to improve their cultivation, exploring the perspective of sustainable food from the sea.

## 1. Introduction

Among the new entries in the sector of food supplements, microalgae have emerged as nutraceutical elements, endowed with beneficial effects on health. Microalgae are a large group of unicellular prokaryotic and eukaryotic organisms that are mainly autotrophic.

Marine microalgae have a high growth/productive rate [1] and interest in these microorganisms from a biotechnological perspective has been rapidly growing in the last decade as a source of sustainable chemical compounds, proteins and metabolites. Some species, such as *Chlamydomonas reinhardtii*, are already recognized as possible industrial biotechnological platforms for the production of a variety of bioactive compounds [2] and are currently subject to genetic manipulations [3,4] with the aim to increase its biotechnological applications.

Microalgae also represent promising opportunities in the field of functional foods due to their production of valuable bioactive ingredients, such as carotenoids with already known health benefits [5], as well other antioxidant compounds [6]. This feature represents an added plus-value since synergism between different families of bioactive compounds strongly enhances their beneficial effects on health [7,8]. Marine microalgae are currently being employed as functional foods due to their high content of polyunsaturated fatty acids (e.g., PUFAs n3 and n6), essential amino acids (e.g., leucine, isoleucine and valine) and pigments (e.g., lutein and β-carotene) and vitamins (e.g., B12). Recently, microalgal biomass has been proposed to enrich wheat flour to produce pasta and related products or to be used as supplements in ready-to-use dry milk products (e.g., baby soups and high protein beverages) that are now employing soybean as the major ingredient [9,10].

The term “nutraceutical” arises from the integration of “nutrition” and “pharmaceutical” and refers to substances related to nutrition, endowed with physiological benefits associated with prevention and/or protection against chronic diseases. Nutraceuticals include mixtures and pure compounds isolated from herbs, as well as dietary derivatives, transformed foods such as cereals, spices, condiments and beverages with beneficial effects on health.

Given their recent classification, there are no specific regulations that classify nutraceuticals as a separate category from drugs, food ingredients or dietary supplements, depending on the country. Nutraceuticals, similarly to dietary regimen and diet, can be used to improve health, prevent chronic diseases, increase life expectancy, delay aging processes or provide structural/functional support to the body. A functional food is defined as food that can deliver additional or enhanced benefits over and above its basic nutritional value. This additional value is often related to health promotion or disease prevention, by adding new ingredients or more of the existing ones. Functional foods may be “designed” to have physiological benefits and/or reduce the risk of chronic diseases beyond basic nutritional functions, may be similar in appearance to conventional food and be consumed as part of a routine diet, as defined by the US Department of Agriculture, Agricultural Research Service [11]. The International College of Nutrition and many other agencies define functional foods as those foods containing some nutrients or bioactive compounds that can have as targets some physiological mechanisms of our body, providing benefits [12]. In recent years, a flourishing market has developed, with increasing request and consumption of supplements and nutraceuticals sold in pharmacies or drugstores.

Here, we explore the current roles of marine microalgae as proposed ingredients for functional foods endowed with potential chemopreventive properties. We analyze the state of the art on the advantages of employing marine microalgae in the diet, focusing on potential chemoprevention applications. We report here an overview of known bioactive molecules, exploring exploited and unexploited microalgal biodiversity, and discussing further developments of microalgal biotechnology in terms of nutraceuticals and functional ingredients for health benefit and disease prevention.

## 2. A Case for Dietary Chemoprevention

Chronic non-communicable diseases (NCD), a characterizing feature of lifestyle diseases, include pathological conditions such as chronic respiratory diseases, metabolic syndrome, type 2 diabetes, cardiovascular diseases and cancer [13]. It has been extensively documented that poor dietary habits, smoking or alcohol abuse, as well as sedentary lifestyle, are the main environmental factors responsible for the early development of these complex diseases [14]. Therefore, the change in food choices/consumption and lifestyle could be very effective in terms of prophylaxis. The USA Dietary Guidelines 2015–2020 (health.gov) exhaustively stated “Shift to healthier food and beverage choices. Choose nutrient-dense foods and beverages across and within all food groups in place of less healthy choices“ [15].

The World Health Organization (WHO) indicates that lifestyle diseases are one of the most serious challenges of XXI century medicine. Statistics from WHO regarding 2016 showed that, among 56.9 million (mln) of total premature deaths, 40.5 mln (71%) were due to noncommunicable diseases (NCDs), i.e., lifestyle diseases, that constituted 7 mln more deaths than in 2000 (+17%) [13]. It has been estimated, that if this growth rate is maintained, then in 2030, the toll might reach 52 mln deaths. According to WHO data, of all deaths caused by lifestyle diseases, 44% (17.9 mln deaths) are due to cardiovascular pathologies, 22% (9 mln) to cancer, 9% (3.8 mln) to chronic respiratory diseases including asthma and chronic obstructive pulmonary diseases and 3% (1.6 mln) to type 2 diabetes [13].

There is a considerable focus on the development of value-added functional food products to combat various diseases like obesity, diabetes, neurodegenerative, anemia, and other chronic diseases [16]. It is believed that, besides metabolic syndrome and obesity, the occurrence of two other of the major pathological conditions, cardiovascular disease and cancer, are also strictly dependent on dietary and lifestyle factors. According to WHO, cancer is, globally, the second leading cause of death and is responsible for an estimated 9.6 mln deaths in 2018; thus, about one of six deaths is due to cancer [17]. We know that many tumors (almost 30%) [18] can be prevented by lifestyle changes, avoiding risk factors such as tobacco use, excessive ultraviolet (UV) exposure, infectious agents, poor dietary habits, lack of exercise, overweight and obesity. Behavioral studies suggest that the promotion of healthy dietary habits and exercising belong to successful strategies.

Epidemiological studies, such as the EPIC cohort [19], provided strong evidence of health benefits from dietary regimens. The shortest survival after diagnosis of cancer is in citizens of Eastern European countries, while those of Mediterranean regions have longer survival rates [20,21]. One of the most beneficial dietary habits to diminish cancer incidence and mortality, except for better medical care, is the Mediterranean diet, which is believed to be one of the healthiest and best-balanced diets in the world [22]. Nevertheless, dietary prevention of cancer is still at its infancy, and we lack deep knowledge of the molecular rationale and on the unveiling of molecular targets.

Chemopreventive molecules are defined as agents able to prevent or delay cancer onset [23]. The concept of chemoprevention is very broad, covering the so-called primary, secondary and tertiary prevention of cancer development and progression, as well as protection from toxic agents, supporting the immune system and the chemotherapeutic effect of drugs. However, cancer prevention still remains a challenging task.

Research on developing innovative options/tools for health benefits and oncological prevention and/or interception [24,25] are of primary importance for society, through investigations on new sources of bioactive compounds. Of all the cancer-hindering factors, dietary intervention and supplements stimulate strong interest in the lay population. Nutraceuticals and functional foods could help oncology through diverse actions [25]. For large access to dietary interventions, functional foods or nutraceuticals, those ingredients have to be easily available, produced or extracted through sources and processes that are environmental-friendly. Dietary interventions, functional foods and nutraceuticals should be envisaged both as preventive measures against cancer and as improving other health conditions.

## 3. Marine Microalgal Biomass and Diversity: A Blue Biotechnology Pillar 

There are many reports on the biological and ecological features of microalgae, underlying their unique properties for the “blue” biotechnology [26,27,28,29]. Microalgae populate all aquatic ecosystems, ranging from freshwater and brackish waters to oligotrophic marine environments. Research on microalgae constantly highlights the huge biodiversity of this group, its richness in terms of physiological traits, adaptive properties to extremely diverse, and sometimes very hostile, environmental conditions. Microalgae are considered as potential leaders of targeted marine species, to be exploited for their numerous advantages in biotechnology.

Microalgae are mostly microscopic unicellular organisms where all biochemical pools or compartments (photosynthetic, reserves, defense, etc.) are located in the same biological compartment (cell) and all biological processes are strongly interconnected within the same space. Conversely to higher plants, the entire organism can be used as a source of reliable products for biotechnology.

Microalgae have a very rapid proliferation rate, thus representing a powerful asset for renewable and highly quantitative production. In parallel, due to their fast generation time and ability to grow in extreme environments (i.e., resisting high levels of environmental stress) microalgae are probably subjected to the increasing rate of molecular evolution [30,31,32]. Moreover, these microorganisms can be easily manipulated genetically, thus enhancing their interest for industrial purposes [33,34].

Most microalgae are oxygenic autotrophs; they convert light into biochemical energy to synthesize biomass. Light is one of the main driving forces of aquatic photosynthesis, being the most variable environmental parameter at short time/space scales, inducing therefore high photosynthetic rate variation. Along the water column, light intensity rapidly varies from limiting (light intensity is below the level of light required for maximizing photosynthetic rate) to excessive (light intensity is much higher than this level) and damageable through photoinhibition with risks for survival, thus requiring secondary metabolites for defense, protective and repair mechanisms. Since biological processes such as photosynthesis, growth, biomass production, division, defense, and excretion derive from light capture, it becomes easy to modulate these processes by manipulating light (intensity and/or spectrum; the so-called “photosynthetic regulation biotechnology”) [1,6,35,36].

Microalgal cells continuously and quickly adapt themselves to ecosystem variations, employing the resources offered by the external environment. Generally, this process involves defense strategy, optimization to cope with environmental stress and lowering energy losses. The strategy is based on synthesizing bioactive molecules with protective functions, limiting or repairing potential damages from a detrimental environment. These bioactive molecules act as primary and/or secondary sources of metabolites, providing the rationale for a key role within the blue biotechnology and nutraceuticals sectors, with applications for human health.

A relevant aspect concerning microalgal production is that, compared to higher plants, microalgae require much less space to be massively produced [1]. They do not need to cover free green space on land that can be potentially dedicated to other activities, but rather vertical blue space. Many efforts are being dedicated to developing biological or technological improvements aimed at lowering the production costs, to propose the institution of industrial implants for large scale microalgal growth, powered by solar energy in old and obsolete buildings, using previously occupied spaces. Microalgae could potentially occupy a considerable market share in the biotechnological sector, but further studies have to be undertaken to allow them to reach this goal.

## 4. Marine Microalgae: A Multifaceted Treasure from the Sea for Human Health

Microalgae represent an abundant source of different classes of compounds of interest for nutraceuticals/food supplements applications. Here, we summarize and discuss selected relevant bioactive compound families with potential chemopreventive effects present in microalgae, alone or in combinations, as candidates for food supplements (Figure 1).

### 4.1. Proteins, Peptides and Amino Acids

Interest in bioactive peptides from marine sources is growing [40,41]. Microalgae are relevant sources of proteins, peptides and amino acids. For example, some microalgae can contain more than 50% dry weight in terms of proteins [42]. Marine peptides exert several beneficial effects on health, such as antihypertensive, antitumor, antimutagenic, anti-tyrosinase, anticoagulant and antioxidant [43,44,45]. Marine proteins have relevant and exclusive properties on the cardiovascular system, such as antihypertensive and Angiotensin I inhibitory activities [46,47,48].

Microalgal protein pool includes many bioactive peptides, proteins made by 2–20 amino acids able to pass through the cell membrane, and therefore endowing them with a hormone-like role in humans [40]. Antioxidant activity of a small peptide (7.5 µM) of *Chlorella vulgaris* exhibits protective activity on DNA and scavenging effect against cellular damage caused by hydroxyl radicals [49]. In addition, the *Chlorella vulgaris* peptide has gastrointestinal enzyme-resistance, while no cytotoxicity effect was observed on Wi38 cells (human normal fibroblasts) *in vitro* [49]. In a recent study, a glycoprotein extracted from the dinoflagellate *Alexandrium minutum* activated specific cell death (mitophagy) on lung cancer cell line (IC_50_ 0.7 µg mL^−1^), without any toxic effect on normal human cells (Wi38) [50]. Lectins, low molecular proteins involved in several biological processes, such as host-pathogen interactions, cell-cell communication and induction of apoptosis have been found in algae and microalgae, such as cyanophytes [51,52].

Mycosporine-like amino acids (MAAs) synthesized by microalgae are another group of bioactive molecules [53]. MAAs are water soluble, low molecular weight and polar compounds [53] that have been shown to protect cells from UV-induced damage due to the presence of double bonds, providing them with antioxidant properties and photostability [54]. MAAs are used in some photo-aging protective formulations and their efficacy have been tested in several human cell lines, such as B16-F1 murine skin melanoma (IC_50_ 3.2 mg mL^−1^) and human HeLa cervical adenocarcinoma (IC_50_ 0.12 mg mL^−1^) [54].

The organosulfur compound dimethylsulphoniopropionate (DMSP), a metabolite of methionine [55], has been reported to increase its concentration when microalgae are exposed to stress conditions such as UV radiation and nutrient starvation [55,56,57]. In microalgae, DMSP and its enzymatic cleavage product dimethylsulphide (DMS) have been demonstrated to exhibit antioxidant activities [56].

### 4.2. Vitamins

Vitamins are essential for health, being precursors of important enzyme cofactors that are required for essential metabolic functions. Vitamins also display strong antioxidant activity. Since most vitamins cannot be endogenously synthesized in humans, they need to be orally consumed. Microalgae represent an unexplored source of nearly all known vitamins: pro-vitamin A (α- and β-carotene, apocarotenoids), vitamin C (ascorbic acid), vitamin E (tocopherols and tocotrienols) and some vitamins of the B group, such as B1 (thiamine), B2 (riboflavin), B3 (niacin) and B12 (cobalamin). Vitamin D is also present in microalgae [57]. Interestingly, microalgal vitamin content varies with external forcing, such as light (in terms of intensity and spectrum) [6], nutritional status of the culture, growth phase and genotype [58,59,60].

#### 4.2.1. Pro-vitamin A

While algae cannot produce vitamin A, they are able to produce its precursors, α- and β-carotene (pro-vitamin A), that are components of the carotenoid pool of all photosynthetic organisms. Although data are already available on carotenoids in microalgae (see below), there is a paucity of knowledge on apocarotenoids [61]. Apocarotenoid biosynthesis starts with the action of carotenoid cleavage dioxygenases and includes biotechnologically interesting compounds, such as retinol (another pro-vitamin A and the main apocarotenoid produced in algae), which is a structural component of the light-sensitive pigment rhodopsin [61]. Natural and synthetic retinoids have been largely demonstrated to inhibit the growth and development of different types of tumors, such as skin, breast, oral cavity, lung, hepatic, gastrointestinal, prostatic, and bladder cancers [62,63,64,65,66].

#### 4.2.2. Vitamin B12

Vitamin B12 (Cobalamin) is a water-soluble vitamin, present in meat products but absent in plants. Some macroalgae and microalgae can synthesize or contain vitamin B12 [67]. Vitamin B12 deficiency is common among people following strict vegetarian or vegan diets [68]. Even though cobalamin from some microalgae (e.g., *Spirulina*) does not seem to be bioavailable [69], vitamin B12 appears to be bioavailable, and used as food supplements, in some microalgae such as *Chlorella* sp. [70] or *Pleurochrysis carterae* (coccolithophid) [71]. This indicates that further research is required to explore the presence and availability of vitamin B12 in microalgal biodiversity, and not only in microalgae mostly used as nutritional supplements. They could be used to supplement vegetarian and vegan diets. Vitamin B12 can act on DNA repair and histone methylation, and high levels of folate and vitamin B12 is associated with a reduced breast cancer risk (see review [72] and references therein).

#### 4.2.3. Vitamin C (Ascorbic Acid)

Ascorbic acid is a water-soluble vitamin with antioxidant properties, essential for the biosynthesis of many compounds in humans [73]. Vitamin C has been reported as a regulator of Hypoxia-Inducible Factor 1α (HIF1α) [74], a major microenvironmental driver of carcinogenesis and tumor angiogenesis. Vitamin C also has effects on extracellular matrix (ECM), impaction on collagen biosynthesis and deposition [75]. Ascorbic acid is used as a food additive and shows beneficial health effects, including cancer and atherosclerosis prevention. Vitamin C acts as an immunomodulatory agent, for example, for the prevention of severe infections such as tuberculosis [76,77]. Ascorbic acid is present in microalgae that can accumulate high concentrations of this vitamin [58,59,60]. A recent study [6] reported a notable quantity of ascorbic acid in a coastal diatom (*Skeletonema marinoi*), and the possibility to modulate its content with light intensity and spectrum variations.

#### 4.2.4. Vitamin D

Vitamin D can exist in several forms, from D_1_ to D_5_. The two main forms of vitamin D in humans are D_2_ and D_3_ and are involved in calcium absorption and metabolism, a crucial process for bone health and homeostasis. Numerous studies have reported on the health benefits of vitamin D in cancer prevention and anti-neurodegenerative effects [78,79,80]. Although poorly documented, it is known that microalgae can contain vitamins D_2_ and/or D_3_ [81,82], together with provitamin D_3_, which is the reason why fish contain high concentrations of vitamin D [57]. Vitamin D, along with its metabolites, besides their well-known calcium-related functions, has been reported to exert chemoprevention activities, through antiproliferative and immune modulatory effects on tumor cells *in vitro* and growth in vivo [83]. Cancer chemopreventive activities by Vitamin D have been documented to act by blocking cell cycle progression (by increasing the expression of cyclin-dependent protein kinase -CDK- inhibitors p21 and p27) [84], modulating the expression of insulin growth factor (IGF-1) [85], blocking cell proliferation via Wnt/β-catenin-signaling pathways [86], and inducing apoptosis or autophagy [83].

#### 4.2.5. Vitamin E (Tocopherols and Tocotrienols)

Tocopherols and tocotrienols are liposoluble antioxidants, protecting membrane lipids from oxidative damage since they are chain-breaking molecules able to prevent the propagation of lipid peroxidation. Vitamin E blocks the production of reactive oxygen species and lipid peroxidation and is involved in the inhibition of low-density lipoprotein oxidation, a process known to have a role in the development of atherosclerosis [87]. Vitamin E can have a chemoprotective role [88], reducing the risk of pancreatic cancer in mice (200 mg/kg) [89]. The Phosphoinositide 3-Kinase Pathway is involved in the activity of Vitamin E and in the inhibition of prostate cancer cell growth [90]. Vitamin E improves endothelial function and vascular health and reduces vascular damage [87].

Vitamin E is synthesized in many microalgae, such as *Dunaliella tertiolecta*, *Tetraselmis suecica*, *Nannochloropsis oculata*, *Chaetoceros calcitrans* and *Porphyridium cruentum* [91,92,93,94] that can be a valuable source of this vitamin as food supplements, since it has been reported that tocopherol content can be higher in microalgae than in terrestrial plants [95].

### 4.3. Pigments

Algae contain photosynthetic pigments that are usually an integral part of the structure of the chloroplast lamellae, with the function to absorb light. Algal pigments are categorized into three classes: chlorophylls, carotenoids and phycobiliproteins.

#### 4.3.1. Chlorophylls

Chlorophylls (*a*, *b* and *c*) are lipid soluble pigments present in photosystems I and II of almost all photosynthetic organisms.

Chlorophylls can be converted into phaeophytin, pyrophaeophytin or phaeophorbide when vegetable food is processed or during ingestion by humans. A semi-synthetic derivative of chlorophylls, chlorophyllin, is commonly used as a food additive (number E141) for its green colour, but the presence of copper (instead of magnesium in natural chlorophylls and derivatives) can be a problem for consumer health.

Chlorophyll or its derived products are known for their health benefits, due to their antioxidant and therapeutic properties. For instance, McQuistan et al. [96] demonstrated a role of chlorophyll in chemoprotection, being able to avoid dibenzo[def,p]chrysene (DBC)-induced DNA adduct formation when it was used as diet (4000 ppm) for trouts. The beneficial effect of chlorophyll has been also reported in preventing the genotoxicity of 4-nitroquinoline 1-oxide (4NQO) in *Drosophila* [97]. In particular, the genotoxicity of orally given 4NQO (3.8 mg) was suppressed by simultaneous administration of chlorophyll (200 mg). The same kind of activity has been reported in *Drosophila* larvae in another study [98], testing a functional mix of ascorbic acid and chlorophylls *a* and *b*. The combination of these three molecules reduced the genotoxic effects of acrolein in exposed *Drosophila* larvae (25%, 7%5, and 250% of ascorbic acid; 0.5, 1, and 2 μM of chlorophylls *a* and *b*). Chlorophylls and their derivatives demonstrated protective action against oxidation. Chlorophyll from *Sanropus androgynus* displayed an antioxidant activity when administered orally to Wistar rats (for 14 days at 8 and 16 µg mL^−1^) injured with the prooxidant sodium nitrate compound revealing a strong in vivo antioxidant activity of chlorophylls [99]. Chlorophylls (in combination with lutein, 50–100 mg mL^−1^) were also able to induce necrosis or apoptosis in hepatoma cells (Hep3B), through the arrest of cells in the G_0_/G_1_ phase [100].

Chlorophyll *a* and a mixture of chlorophylls *a* and *b* display different functional activities related to health benefits; in fact, these molecules displayed chemopreventive effects, such as antioxidant activity, inhibition of cytochrome P450 enzymes, increased level of glutathione S-transferase, inhibition of cell differentiation, promotion of cell arrest and apoptosis [101]. Given the high content of chlorophyll per microalgal biomass unit (0.5–1 pg cell^−1^), at times more abundant than in higher plants [40], microalgae are a highly promising source for the extraction and production of these molecules. For chlorophyll *c*, present in “brown algae” (and not in green algae or plants) knowledge on its health benefit is lacking. This gap of knowledge requires further investigation since microalgae and in particular diatoms have a huge biotechnological potential [1].

#### 4.3.2. Carotenoids

Carotenoids are accessory pigments mainly involved in light-harvesting processes. Some of them act as structural stabilizers for protein assembly and can have a role in photoprotection against light damage [102].

Carotenoids are hydrophobic highly conjugated 40-carbon (with up to 15 conjugated double bonds) molecules. More than 500 species of natural carotenoids are known and classified in two groups: carotenes (hydrocarbon carotenoid) and xanthophylls (oxygenated derivatives of carotenes).

The interest of microalgae as carotenoid sources are due to (i) the high biodiversity of carotenoid species, (ii) the high content of carotenes and xanthophylls per cell, (iii) the great capacity of microalgae to modulate the content of carotenoids.

Numerous algal carotenoids are also found in terrestrial plants (β-carotene, zeaxanthin, antheraxantin, violaxanthin, neoxanthin and lutein). Other carotenoids are specifically present in algae, such as astaxanthin, loroxanthin, fucoxanthin, diadinoxanthin, diatoxanthin, siphonein. Health benefits are mainly known for those carotenoids present both in terrestrial plants and algae (mainly green algae), while it remains to be discovered the mechanism of action of the carotenoid species present exclusively in algae. β-carotene is known to be able to scavenge harmful oxygen and nitrogen radicals [103], and it is the main form of provitamin A (for more details and for the other forms, see Section 4.2.1 on pro-vitamin A). For its healthy properties, β-carotene has many applications in food, pharmaceuticals and cosmetics. β-carotene has protective effects against UV and oxidative damage [104], and it has been demonstrated that oral intake of β-carotene (from 30 to 180 mg d^−1^) can prevent UV-induced erythema in humans [105,106]. It also displays protective effects against atherosclerosis [107] and improves retinal and visual dysfunctions [108]. β-carotene plasma levels were inversely connected with atherosclerotic lesions, inhibiting low-density lipoprotein oxidation and influencing plasma triglycerides, cholesterol and high-density lipoprotein [109]. A role of β-carotene in lowering the rate of several types of cancer and degenerative diseases in humans (20 mg d^−1^) and maintaining their average serum levels has been discussed [110].

Lutein and zeaxanthin, present both in terrestrial plants and green algae, are essential pigments in the *macula lutea* of the human eye retina. Lutein protects photoreceptors by filtering blue light and can reduce by 40% light incidence that damages the retina [111,112]. This pigment is receiving interest for human health, given its ability as free radical scavenger and capacity to quench singlet oxygen [111,112]. Lutein, in association with zeaxanthin (considering a daily intake of 6 mg in humans), protects tissues from free radicals and can prevent atherosclerosis, cataract, diabetic retinopathy and age-related retinal degeneration [113,114]. Lutein also possesses anticancer activity [115] and protects endothelial cells *in vitro* [116]. A mix of carotenoids, composed by lutein, violaxanthin, antheraxanthin, neoxanthin and carotenes (100 µg mL^−1^ of hydroalcoholic extract of the green microalga *Tetraselmis suecica*) act with synergistic effects on the activation of defense mechanisms in human cells injured by exposure to an oxidative agent (30 mM of H_2_O_2_) [117].

Zeaxanthin is a photoprotective pigment in plants and green algae [102] and presents enhanced antioxidant activity compared to other higher plant xanthophylls [118]. Like lutein, zeaxanthin has a protective role and helps in eye health [119] and improving vision [120]. An in vivo study [121] demonstrated that an 8-weeks period of lutein and zeaxanthin intake (7.3 mg of meso-zeaxanthin, 3.7 mg of lutein and 0.8 mg of zeaxanthin) produced an enhancement of macular pigments in humans. Little information is available on zeaxanthin or lutein in chemoprevention but they present great potential. Another xanthophyll, violaxanthin, extracted from *Dunaliella tertiolecta*, is known to have antiproliferative activity [122]. The study showed dose-dependent growth inhibition of MCF-7 (human breast adenocarcinoma cell line) exposed to 0.17 μM of enriched extract.

Astaxanthin, an aquatic life carotenoid, is present in some microalgae [123] and aquatic animals [124]. It displays a great antioxidant activity, almost 100 times higher than α-tocopherol, protecting cells against oxidative damage and lipids from peroxidation [125]. For this reason, it has many applications in nutraceuticals, cosmetics and food products, is well known as a human dietary supplement. Astaxanthin has protective effects against diseases such as cancer, inflammation, metabolic syndrome, diabetes, neurodegenerative and eye diseases [126]. The intake of astaxanthin is known to repress heart diseases, affecting cholesterol levels in blood and preventing obesity. A clinical study describes astaxanthin administered to 61 non-obese individuals (6 to 18 mg d^−1^) influence triglyceride and HDL-C in relation to an increase of adiponectin [127].

Fucoxanthin, a marine life xanthophyll is present in some microalgae (e.g., diatoms) and brown macroalgae. Strong interest in this pigment has developed in recent years for its role in anti-obesity by promoting the oxidation of fatty acids and heat production [128]. Fucoxanthin has a great antioxidant activity as well as anti-inflammatory, antidiabetic, antiphotoaging and neuroprotective properties [129,130,131]. Moreover, fucoxanthin displays anticancer activity in several experimental models against a variety of cancer types including colon cancer (HT-29 cells, 15 µM of fucoxanthin) and leukemia (HD-60 cells, 10 to 45 µM of fucoxanthin) [132].

Among the numerous other pigments with unknown health benefits, diadinoxanthin and/or diatoxanthin are probably of great interest. These xanthophylls are mainly present in chlorophyll *c*-containing algae and have the same physiological role as violaxanthin and zeaxanthin in green algae or plants. Studies on diatoms proved an efficient antioxidant role of these pigments in their own photoprotection [133].

#### 4.3.3. Phycobiliproteins

Phycobiliproteins are a family of proteins present in algae belonging to the cyanobacteria (e.g., *Spirulina*), red algae (e.g., *Porphyridium*, *Rhodella*, *Galdieria*), cryptophyta and glaucophyta [134]. They represent a family of highly soluble and stable fluorescent proteins. These proteins contain covalently linked tetrapyrrole groups, that play a biological role in collecting light. They have a linear tetrapyrrole prosthetic group (bilins) and adsorb wavelengths between 470 and 660 nm [134]. Four groups of phycobiliproteins are known: phycoerythrin, phycoerythrocyanin, phycocyanin and allophycocyanin. Phycobiliproteins are already used in biotechnological applications, such as in food and cosmetic production. The well-known and integrated health benefit of *Spirulina* also relies on the high presence of phycocyanin [135]. Indeed, health benefits from phycocyanin have been described [136], for instance as antioxidant [137], anti-inflammatory, hepatoprotective, neuroprotective, free radical scavenger, immunomodulation and atheroprotective [134,138,139,140,141].

Due to their richness in phycobiliproteins, as well as in zeaxanthin and other bioactive compounds, investigations and cultivation of other marine cyanophytes (other than *Spirulina* or *Nostoc*) is a challenge due to biotechnological issues linked to their production and an important goal for health protection.

### 4.4. Fatty Acids

Fatty acids (FA) are classified as saturated (SFA), monounsaturated (MUFA) or polyunsaturated (PUFA) on the basis of the number of double bonds. PUFA’s are classified in omega 3 and 6 (ω3 and ω6), depending on the position of the first double bond in relation to the terminal methyl end of the carbon chain. Fatty acids are structural components of phospholipids, glycolipids, and triacylglycerols (TAGs) [142], and all living organisms require a large pool of FA for health and survival. Humans are not able to endogenously synthesize PUFA’s exceeding 18 carbon atoms. Since they are known to provide benefits for human health, their intake requires external administration, by diet [142]. Recent studies showed that fishes and higher plants do not synthesize these fatty acids de novo, but accumulate eicosapentaenoic acid (EPA, 20:5) and docosahexaenoic acid (DHA, 22:6), two PUFA’s with beneficial properties for human health. Fishes accumulate PUFA’s by feeding on microalgae [143], that are rich in lipids and fatty acids, although the type and the amount vary considerably with phylogeny and in relation to external forcing (environmental or culture) conditions [144]. Microalgae are a source of long-chain PUFAs especially of the ω6 family, such as γ-linoleic acid (GLA, 18:3), arachidonic acid (AA, 20:4), ω3 family as EPA and DHA [142,144].

Recent studies highlighted the anti-inflammatory properties of EPA and DHA, together with an improvement of cardiovascular health [142]. The dietary supplementation of EPA and DHA (2 g day^−1^) suppresses *in vitro* activation of TLR4, IL6 and IL8 in adipose and trophoblast cells isolated from adipose tissue and placenta of treated pregnant women [145]. Microalgal derived PUFA’s have a role in cognitive and brain development [146,147] and against brain-related diseases [148,149]. The ω3 and ω6 fatty acids have attracted attention for their potential anticancer effects [21].

### 4.5. Phytosterols

Sterols are amphipathic compounds that originate from isoprenoid biosynthesis, forming a group of triterpenes with a tetracyclic cyclopenta [a]-phenanthrene structure and a long flexible side chain at the C-17 [150]. Sterols are components of cellular membrane acting on its stability and fluidity. Cholesterol is the most abundant sterol in animals while β-sitosterol is the main contributor of phytosterols [151].

Microalgae contain an elevated concentration of sterols [150,152] and these amphipathic compounds reach 5.1% of the total biomass in the microalga *Pavlova luteri* [152]. Moreover, microalgae display a high diversity of phytosterols, such as brassicasterol, sitosterol, and stigmasterol [150,153]. Some microalgal species (such as *Pavlova sp*., *Tetraselmis sp*. and *Nannochloropsis sp*.) contain a mixture of ten or even more phytosterols [153]. Sterol composition varies depending on the algae strain and can be modulated by light intensity, temperature or growth phase [150]. All these properties lead microalgae to be a promising source, of phytosterols (mainly unknown), for health benefits.

Phytosterols are suggested as dietary supplements for lowering cholesterol and decreasing the number of cardiovascular diseases [154]. Phytosterols can act as secondary messengers in a hormone-like manner, leading to relevant cellular processes such as development and neurotransmission [155].

Beneficial effects of microalgal-derived phytosterols were experimentally reported, such as anti-cancer, anti-inflammatory, antioxidant or anti-cholesteroligenic properties [156,157]. Anti-metastatic activity was reported for fucosterol, a brown algal phytosterol [157,158]. Fucosterol and isofucosterol are also present in microalgal species (*Chrysoderma* sp., *Chrysomeris* sp., *Chrysowaernella sp*. and *Giraudyopsis sp*. [159]), however, their bioactivity has not yet been elucidated.

### 4.6. Polysaccharides

Polysaccharides are polymeric carbohydrates, large hydrophilic molecules, with fundamental roles in living organisms, such as energy storage, protection and structural molecules [160]. Simple carbohydrates are produced by photosynthesis. Polysaccharides have many biotechnological applications, including the food industry. These compounds are present in high concentration and diversity in microalgae [161]. Microalgae are able to accumulate carbohydrates to more than 50% of their dry weight [162], due to the high photoconversion efficiency of the photosynthetic process. Microalgal carbohydrates can be found under the form of starch, glucose, sugar or other polysaccharides.

Healthy beneficial effects of polysaccharides are known, and include boosting of the immune system [160,161,162,163], or anti-viral properties [164], or blockade of tumorigenesis [165]. In particular, crude polysaccharide extracts of *Chlorella stigmatophora* (IC_50_ 2.25 mg kg^−1^) and *Phaeodactylum tricornutum* (IC_50_ 2.92 mg kg^−1^) showed a high anti-inflammatory effect on carrageenan-induced paw oedema in rats, higher than the reference drug used (indomethacin, IC_50_ 8.58 mg kg^−1^) [163]. The immuno-stimulating effect was observed for the high-sulphate-containing exopolysaccharide p-KG03, which is produced by the red-tide microalga *Gyrodinium impudicum*. Yim and collaborators [165] highlighted how protective functions mediated by cells of the immune system (such as macrophages and natural killer) were enhanced by treatment with p-KG03 in vivo, when mice were treated with a single dose of this polysaccharide (100 or 200 mg kg^−1^). This stimulation produced an increase in tumoricidal activity of macrophages and natural killer cells [165].

Antioxidant properties of polysaccharides are also known. For instance, crude extracellular polysaccharides (from 0.4 to 1.6 mg mL^−1^) from the red unicellular alga, *Rhodella reticulata* resulted in antioxidant ability higher than tocopherol [166]. Polysaccharides also have strong in vivo anti-triglyceride and cholesterol roles [167]. Animals fed with *Rhodella reticulata*, a red alga rich in polysaccharides, lowered levels of cholesterol in the serum, with consequent reduction of insulin and glucose. Gardeva et al. [168] described the anti-proliferative activity of polysaccharides of the alga *Porphyridium cruentum*. In particular, 10 mg kg^−1^ of extract was able to reduce in vivo myeloid tumor growth in hamsters. Yet, the primary culture of these tumor cells showed a high percentage of mortality when treated with 100 µg mL^−1^ of polysaccharide extract.

### 4.7. Polyphenols

Polyphenols are divided into phenolic acids, flavonoids, isoflavonoids, stilbenes, lignans, and phenolic polymers [169]. Phenolic compounds are composed by one or more aromatic rings with one or more hydroxyl groups. Polyphenol intake in humans comes from fruits and vegetables [169], but also from tea, wine or coffee. Generally, phenolic acids account for one-third of the total intake and flavonoids for the remaining two thirds [169]. In addition to their antioxidant activity, these compounds display a wide range of biological activities such as anti-inflammatory, anti-cancer, anti-allergic, anti-diabetes, anti-aging and antimicrobial properties [170].

In algae, these compounds are mostly involved in protective activities against external factors, such as herbivorous grazing, UV radiation, metal contamination or light [6]. Extensive information on polyphenols exists in macroalgae [171]. Phenolic acids and flavonoids have been revealed also in diverse microalgae (*Ankistrodesmus* sp., *Spirogyra* sp., *Euglena cantabrica*, and *Caespitella pascheri*) and cyanophytes (*Nostoc* sp., *Nostoc commune*, *Nodularia spumigena*, *Leptolyngbya protospira*, *Phormidiochaete* sp., and *Arthrospira platensis*), highlighting their high variability among strains [172]. The high amount of phenolic acids was observed in two different microalgal species: the green alga *Dunaliella tertiolecta* and the diatom *Phaeodactylum tricornutum* [173,174]. Moreover, it has been observed that several microalgal species have total phenolic content similar to or higher than several fruits and vegetables [169,170].

One intriguing example of polyphenols produced by some microalgae is represented by the marennin [175], which possesses interesting bioactivities: antioxidant, antibacterial, antiviral, and inhibitory effects on tumor cell growth (melanoma, lung and kidney carcinoma [176]). This polyphenol produced by the diatom *Haslea ostrearia* is blue and is responsible for the greening of oysters in western France [177]. In addition, for this class of compounds, further studies are needed to deeply explore their production in other microalgae.

## 5. Linking Past and Future: Challenges for Microalgae Ingredients in Nutraceuticals

Microalgae have become optimal candidates as sources of natural antioxidants, as revealed by a number of studies [178,179]. The role of microalgae as nutraceutical additives dates back to thousands of years [180]. Chinese populations used the cyanophyte *Nostoc* 2000 years ago to survive during famine [181]. Another cyanophyte, *Spirulina* harvested from lakes, is known to have been used by the Aztecs in the sixteenth century [182]. The same species is reported as the food of the Kanembu population living along the shores of Lake Chad, Africa [183,184]. *Spirulina* is probably the emblem of using microalga for improving human benefits and helping human survival or wellness (the so-called “wonderful future food source” [182]). This is due to its exceptionally high protein content with balanced essential amino acids content [182]. Until now, few microalgae are used as a source of additives for human or animal food [185], and among them, the greater contribution comes from *Chlorella vulgaris* and *Spirulina pacifica* [186].

The special niche occupied by *Spirulina* in biotechnology has also been invaded in recent years by other species (*Chlorella, Haematococcus*, *Dunaliella, Tetraselmis*), and the biomass production of microalgae has been reconsidered for varied biotechnological applications [10]. Due to the richness of microalgal biodiversity, and the high concentrations of bioactive molecules in cells, one of the aspects to be enhanced is a screening of microalgal biodiversity, exploring the functional traits of algae, that regulate their potential to biochemically modulate and thus synthesize bioactive molecules. The advantage of microalga is the presence of a plethora of bioactive molecules in the cell (see above), together with high species- and environment-dependent concentration of primary and secondary metabolites of interest.

Challenges consist of a full screen of bioactive molecules synthesized by microalgae to cope with environmental stress. Some of them are relatively well known (e.g., carotenoids-xanthophylls, phycobiliproteins [187]). Others are less known or information on their bioactivity is very partial. Once quantitative and qualitative information will be available on the beneficial effects of bioactive pools in microalgae, research on their molecular pathways as well as on their interactions has to be advanced. In parallel, their function in cells, as well as their cellular localization and roles, have to be elucidated.

To be an efficient and promising functional beverage/food ingredient, a microalga needs to provide bioactivity to the food, presenting a high antioxidant activity and/or ability to activate biological processes of defense, repair, and immunological responses. A tool to enrich food or beverages is using a miscellaneous mixture of different compounds from different microalgae. For instance, mixing lipophilic and hydrophilic bioactive compounds found in microalgae might be a route to be pursued for enhancing the beneficial effect of foods/beverages. Other aspects to be further improved concern the taste of the functional foods/beverages and dietary supplements from microalgae as well as the maintaining of bioactivity capacity with time during conservation.

## Figures and Tables

**Figure 1 nutrients-11-01226-f001:**
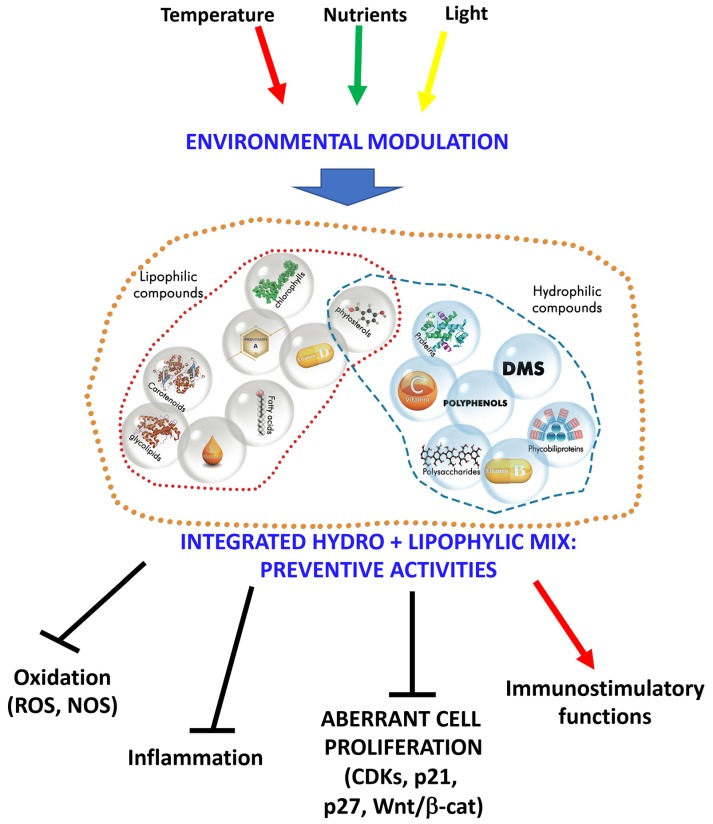
Diversity of bioactive compound families with potential chemopreventive effects present in microalgae as a candidate for food supplements. The hydrophilic and lipophilic compounds’ groups are distinct. In order to provide a wide range of protective effects, the synergy between groups of molecules is required. This can be obtained with a two-step process (mixing the two hydro- and lipophilic extracts [37,38] or in one step process [39]. The combinations proposed will result in enhanced preventive activities action by blocking oxygen (ROS)/nitrogen (NOS) reactive species generation, inflammation, aberrant cell proliferation (such as cancers) and by potentiating the activities of the immune system (immunostimulatory functions). DMS: dimethylsulfide, ROS: oxygen reactive species; NOS: nitrogen reactive species; CDKs: cyclin-dependent-kinases.

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
