# Peer review of "Microalgal Derivatives as Potential Nutraceutical and Food Supplements for Human Health: A Focus on Cancer Prevention and Interception"

_nutrients, 2019, doi:10.3390/nu11061226_

Reviewer 1 Report

This review paper sums up the latest discoveries of applied microalgal research, and focuses especially on microalgae as a source of functional ingredients for food and nutraceuticals. I find the manuscript well written and comprehensive. I have no other suggestions, but encourage minor English spell check.

Author Response

REVIEWER1

This review paper sums up the latest discoveries of applied microalgal research and focuses especially on microalgae as a source of functional ingredients for food and nutraceuticals. I find the manuscript well written and comprehensive. I have no other suggestions but encourage minor English spell check.

Response: We thank the reviewer for her/his comments, and we have now improved English spelling.

Reviewer 2 Report

In this review, authors have analyzed the state of the art on the advantages of employing marine microalgae in the diet, focusing on potential chemoprevention applications. They report an overview on known bioactive molecules and have discussed microalgal biotechnology in terms of nutraceuticals and functional ingredients for health benefit and disease prevention.

Many references can be found in bibliography about potential nutraceutical compounds from microalgae, but this review has updated information and new references. The paper is well written, understandable, with updated references and information.

Author Response

REVIEWER2

In this review, authors have analyzed the state of the art on the advantages of employing marine microalgae in the diet, focusing on potential chemoprevention applications. They report an overview on known bioactive molecules and have discussed microalgal biotechnology in terms of nutraceuticals and functional ingredients for health benefit and disease prevention. Many references can be found in bibliography about potential nutraceutical compounds from microalgae, but this review has updated information and new references. The paper is well written, understandable, with updated references and information.

Response: We thank reviewer for her/his kind comments.

Reviewer 3 Report

The review proposed by Galasso et al. entitled "Microalgal Derivatives as Potential Nutraceutical and Food Supplements for Human Health: a Focus on Cancer Prevention and Interception" is, in general, well written but lacks originality. A revision of the manuscript addressing the above mentioned concerns is needed before the manuscript is accepted for publication :

1.The defenition of microalgae is not complete and comes very late in the text. A modification in this sens should be done.

2.There is a lack in illustrations, no tables as well, and the authors proposed only one figure which is not not relevant at all at this state.

3. There is no information regarding the potential use of Chlamydomonas reinhardtii which considering as a biological and biotechnological model.

4. It's very important that authors update the regulatory directives concerning the use of microalgae as novel food source section where the  most recent reference is from 2014.

Author Response

REVIEWER3

1) The review proposed by Galasso et al. entitled "Microalgal Derivatives as Potential Nutraceutical and Food Supplements for Human Health: a Focus on Cancer Prevention and Interception" is, in general, well written but lacks originality. A revision of the manuscript addressing the above-mentioned concerns is needed before the manuscript is accepted for publication:

Response: Our review aimed to report and integrate previous studies on microalgae for health-protection focusing on their applications as nutraceuticals for cancer prevention. Due to the enormous number of compounds that can be exploited, we hardly selected the information reported in the text for each family of compounds to limit the length of the paper (already long enough!). Our review has the advantage to put together many different families of compounds that are often treated separately, highlighting the interests in integrating them in new functional foods. Moreover, some families or some compounds synthetized by microalgae (e.g., vitamins D or B, DMSO, MAAs, etc.) are poorly documented and are still not the object of an integrative review. In conclusion, the strength of this paper is based on the integration of information on the chemoprevention-oriented bioactivities of the different compounds/families, together with their richness in microalgal biodiversity.

2) The defenition of microalgae is not complete and comes very late in the text. A modification in this sens should be done.

Response: We acknowledge the reviewer for his/her pertinent remark. We change the text accordingly, introducing microalgae in the section 1, lines 45-56.

3) There is a lack in illustrations, no tables as well, and the authors proposed only one figure which is not relevant at all at this state.

Response: Figure 1 provides the rational to employ the integrated hydro+lyphophylic mixture, based on the relevant cellular and molecular targets discussed in our review. Therefore, we believe this figure is necessary to point out the originality of our work. We feel that no table is necessary, since our work exhaustively illustrate the relevant and major insights within the current and updated literature"

4) There is no information regarding the potential use of Chlamydomonas reinhardtii which considering as a biological and biotechnological model

Response: We added some information on Chlamydomonas in this revised version (see introduction), retrieved from three recent publications highlighting the interest of this species for biotechnological applications.

5) It's very important that authors update the regulatory directives concerning the use of microalgae as novel food source section where the most recent reference is from 2014.

Response: We thank the reviewer for the suggestion, the first authorization by EU was released in 2014, but the last addition in terms of microalgae species for novel food was recently updated (July 2018). We have added this information in the text of the revised version.